# Relative Deprivation Leads to the Endorsement of “Anti-Chicken Soup” in China

**DOI:** 10.3390/ijerph192114210

**Published:** 2022-10-31

**Authors:** Xiaomeng Zhang, Tianxin Wang, Zhenzhen Liu, Xiaomin Sun, Shuting Yang

**Affiliations:** 1Education and Human Development, Nagoya University, Nagoya 464-8601, Japan; 2School of Philosophy, Psychology & Language Sciences, College of Arts, Humanities & Social Sciences, The University of Edinburgh, Edinburgh EH8 9YL, UK; 3Beijing Key Laboratory of Applied Experimental Psychology, National Demonstration Center for Experimental Psychology Education (Beijing Normal University), Faculty of Psychology, Beijing Normal University, Beijing 100875, China

**Keywords:** endorsement of anti-chicken soup, relative deprivation, income inequality, self-handicapping

## Abstract

“Anti-chicken soup” (ACS) persuades people to yield to reality and give up rather than encouraging people to work hard as “chicken soup” does. The current study explored whether people with a higher level of relative deprivation (RD) would be more likely to endorse ACS. We found that people with high-measured (Study 1) and manipulated (Study 2) RD were more likely to endorse ACS. Study 2 also suggested that the effect was mediated by self-handicapping. It seems that relatively deprived individuals may adopt the strategy of self-handicapping so that they could attribute their failure to external causes, which in turn leads to lower motivation to try their best and ultimately the endorsement of ACS.

## 1. Introduction

In 2016, a screenshot of a skinny, middle-aged man, the Chinese comedian Ge You, slouched on a couch lazily and mindlessly went viral on Chinese social media. Chinese netizens coined the phrase “Ge You-esque slouching” to describe a state of “living without hope”. Originally taken from a popular TV show, the photo spawned a wave of “anti-chicken soup” (ACS) in China. Different from “chicken soup” (CS), which can be traced back to Jack Canfield’s “Chicken Soup for the Soul” aiming at inspiring and encouraging people to work hard and try their best to fulfill their dreams, ACS goes in the opposite direction. It is full of negative energy [1,2] that emphasizes the cruelty of reality and persuades people to yield to their fate rather than devote every effort to gaining success. The typical ACS motto, “Nothing is difficult for the man who will give up”, is obviously an ironic mock of the famous CS: “Nothing is difficult for the man who will try.” Triggered by the “Ge You-esque slouching”, ACS is so popular in society that, in 2017, a new brand of drink, “Sang Tea” first came out in Shanghai, and it quickly became popular among consumers around the country. In Chinese, “Sang” means sad, downcast, and discouraging. Consistent with its name, every cup of Sang tea was printed with a sentence advocating ACS. For example, “The problems you can solve with money are not problems, but how to get money is your biggest problem” and “Everything is hard in the beginning, hard in the middle, and even harder in the end”. People also call it Nega-tea-vity.

ACS has been described by Chinese state media, People.cn, as harmful content that erodes youth spirits [3]. In addition, lots of researchers have expressed concern about this phenomenon. Endorsement with such negative information is believed to magnify the disappointing aspects of life [4], lead to people’s pessimistic judgment of the world and themselves [5], and affect their daily life and work [6]. To diminish the negative consequences of the endorsement of ACS, it is of great importance to investigate what factors will affect the endorsement of ACS. However, to the best of our knowledge, such research is scarce. Therefore, the current research focused on relative deprivation and aimed to examine the effect of relative deprivation on the endorsement of ACS as well as its psychological mechanism.

### 1.1. Income Inequality and Relative Deprivation

To understand the popularity of ACS, it is necessary to put it into the social context of increasing economic inequality. According to the China Private Wealth Report 2017, in China, 0.1% of the population owns 30% of the country’s individual investable assets. The China Statistical Yearbook published by the National Bureau of Statistics of China shows that from 2010 to 2016, China’s Gini coefficient was above 0.46. Generally speaking, when the Gini coefficient is between 0.4 and 0.5, it means that the income gap in the region is very large, and if the Gini coefficient is greater than 0.5, it means that the income gap in the region is very large. It indicates severe economic inequality in China. At the same time, the convenience of social networking makes the daily life of those people standing at the top of the social ladder more accessible. In this context, people are likely to feel relatively deprived. According to Smith et al. (2012), RD occurs when an individual compares themselves with others, perceives that he/she is in a disadvantaged situation, believes that it is unfair, and he/she feels resentment [7]. Marx (1847) vividly depicted the occurrence of RD in his writing: “A house may be large or small; as long as the neighboring houses are likewise small, it satisfies all social requirements for a residence. However, let there arise next to the little house a palace, and the little house shrinks to a hut” (p. 33) [8]. Inequality makes people feel that they are less well-off relative to those above them on the social ladder. The steeper the ladder is, the worse people’s self-evaluation of their own status. Hastings (2019) summarized that such chronic negative evaluations would have an impact on people’s attitudes and behaviors regarding work, family, and many other aspects of their lives [9].

### 1.2. Relative Deprivation, Endorsement of Anti-Chicken Soup, and Self-Handicapping

The current research proposed that relative deprivation may be an important factor that induces people to endorse ACS. On the one hand, endorsement of ACS embodies the tendency of people to surrender to their fate after being hit by a cruel reality rather than devote every effort to gain success, which is similar to withdrawal behaviors or escape behaviors. Evidence in extant literature indicated that those who experience relative deprivation are more likely to conduct withdrawal behavior or escape behaviors, such as smoking [10], drinking alcohol [11], gambling [12], and watching television [13]. In addition, studies showed that young people that feel relatively deprived became unwilling to work hard and gave up their chances to achieve their success goals [14,15,16,17] just like ACS persuades people to do. On the other hand, ACS reflects a state of anxiety, hopelessness, discouragement, dissatisfaction, and depression [18]. Emotionally, those who endorse ACS tend to feel lost, helpless, and depressed, which is consistent with the negative consequences of RD. Previous research indicates that relative deprivation positively predicts negative internal states such as pressure, anxiety, depression, despair, and pessimism [7,19,20,21]. Therefore, we assume that people who feel relatively deprived tend to endorse ACS and proposed the following: 

**Hypothesis** **1** **(H1):**
*RD is positively associated with people’s endorsement of ACS.*


Why does relative deprivation positively predict the endorsement of ACS? We proposed that self-handicapping may be the underlying mechanism. Relative deprivation occurs when individuals perceive that they are at a disadvantage situation compared with others. Amidst disadvantaged situations, people need to attribute their own failures either internally or externally. Internal attribution will cause an inevitable threat to people’s self-esteem and confidence in their own worth and abilities. Thus, to avoid such threats, people are likely to choose to rationalize their disadvantage [22], which could be done by externalizing the failure. Such external causality makes people less likely to make efforts on their own for the purpose of improving their inferior position, which refers to self-handicapping [23]. Associated with reducing effort behaviors, self-handicapping is a strategy by which people make excuses in the hopes of keeping potential failure from hurting their self-esteem [24]. By externalizing failure, self-handicapping can protect and maintain self-worth [25]. In most studies related to self-handicapping, researchers regard reducing or abandoning efforts as a sign of self-handicapping. An empirical study has shown that in highly self-related tasks when the expected rate of success is low, participants will deliberately decrease their efforts to reduce the threat of failure and to protect their self-regard [26]. There is also other experimental evidence, e.g., [24,27,28,29,30,31], consistent with these findings. When experiencing relative deprivation, people tend to believe that they are in a disadvantaged situation and such a situation is difficult to ameliorate [32]. Following this vein, it is rational to propose that people who feel relatively deprived may tend to adopt a self-handicapping strategy to cope with one’s disadvantaged situation.

People who adopt the self-handicapping strategy may tend to endorse ACS. People have an inherent tendency to avoid discomfort out of the mismatch between their behaviors and attitudes, which is also known as cognitive dissonance [33]. Analogically, self-handicappers would like to turn their attention to information that matches their self-handicap-state unconsciously [34]. The key feature of ACS is to tell people their disadvantaged states are not due to themselves and to discourage people from working hard. ACS, for instance, urges people to realize the cruel reality by using such questions “How can you know the bitter taste of desperation if you don’t give it a try?”. It tries to convey the idea that it is impossible for you to catch up with those above you no matter how hard you try. This idea fits perfectly with the mental state of self-handicappers and rationalizes their disadvantaged states and tendency to avoid making more effort. Thus, it is natural for them to endorse ACS.

Based on the above argument, living in a period with an unprecedented level of income inequality, people tend to feel relatively deprived. Extreme RD with no hope of catching up to those at the top of the social ladder tends to threaten self-esteem. People are forced to find reasons for their failures without harming their self-evaluation. Self-handicapping is a strategy for individuals to protect their self-esteem. However, the cost of self-handicapping is that it holds people back from making efforts on their own. This conclusion is in line with the gist of ACS, i.e., persuading people from striving. Thus, self-handicappers tend to endorse ACS. Therefore, we proposed the following:

**Hypothesis** **2** **(H2):**
*Self-handicapping mediates the relationship between RD and the endorsement of ACS. Specially, RD was positively related to self-handicapping, which in turn was positively related to the endorsement of ACS.*


In the current research, two studies were conducted to examine these hypotheses by measuring (Study 1) and manipulating (Study 2) RD.

## 2. Study 1

### 2.1. Methods

#### 2.1.1. Participants and Procedure

A total of 236 participants were recruited via a paid online research participation system (www.sojump.com, accessed on 4 March 2019) and passed the attention check item, which stated, “This question is designed to test whether the scale is displayed on your screen accurately. Please choose the answer ‘strongly disagree’ to this question”. Any choice other than the required one was wrong. In this way, those participants who provided their responses carelessly would be identified. Among the participants, 168 were female and 68 were male (Mage = 27.39, SD = 6.57; aged 18 to 49 years old with 33 missing values). Regarding education levels, 7 participants reported high school or below, 15 reported junior college, 176 reported college or university, and 38 reported postgraduate studies or above. For post-tax monthly income, 112 participants reported less than CNY 4000, 85 reported CNY 4000–8000, and 37 reported more than CNY 8000 (with 2 missing values). We performed a sensitivity power analysis using G*Power [35] for the effect of RD on ACS endorsement. A sample of 236 would provide 80% power to detect an effect of *f*^2^ = 0.03 and 90% power to detect an effect of *f*^2^ = 0.04, assuming an alpha of 0.05.

All participants signed an online consent form before they were directed to the online questionnaire. After completing the task, participants received an online debriefing and 10 RMB each in payment for their participation.

#### 2.1.2. Measures

Relative Deprivation. RD was measured with the 5-item Chinese version [36] of the Revised Personal Relative Deprivation Scale [37] (e.g., “When I think about what I have compared to others, I feel deprived”) using a 5-point scale (1 = strongly disagree to 5 = strongly agree). The item scores were averaged, with higher scores indicating a higher level of RD (α = 0.67).

Anti-Chicken Soup Endorsement. To measure individuals’ attitudes toward ACS, we first collected a pool of the most popular ACS mottos in 2018 from Weibo (China’s largest internet social platform) and Zhihu (China’s largest online Q & A community platform). Only those ACS mottos that received more than 200,000 “likes” on the platforms were selected. As a result, 49 mottos entered the preliminary pool (e.g., “How can you know the bitter taste of desperation if you don’t give it a try?”). Then, 42 college students from the first author’s institution (23 males, Mage = 19.90, SD = 3.46) were invited to rate the representativeness of these items. Participants were provided with the definition of ACS and then were asked to rate to what extent they believe that the listed items are in accord with the definition on a 7-point Likert-type scale (1 = definitely no, 7 = definitely yes). Finally, the top-rated 12 mottos (M = 5.21, SD = 1.30) were selected as the final ACS mottos list.

To measure ACS endorsement, participants were asked to rate to what extent they endorsed these 12 items on a 5-point Likert scale ranging from 1 (absolutely disagree) to 5 (absolutely agree). The item scores were averaged, with higher scores indicating a higher level of ACS endorsement (α = 0.81).

Demographic Information. Participants’ gender (male coded as 1; female coded as 0), age, educational level (1 = “high school or below”; 2 = “junior college”; 3 = “college or university”; 4 = “postgraduate studies or above”), and post-tax monthly income with 4 categories (1 = “less than 4000 RMB”; 2 = “4000–8000 RMB”; 3 = “8000–10,000 RMB”; 4 = “more than 10,000 RMB”) were collected and controlled in the data analysis.

#### 2.1.3. Missing Values Analysis

The overall proportion of missing values was 0.77%. Three items of ACS endorsement had missing values and each item had 1 missing value (3 missing values in total, 0.10%). Thirty-three participants did not provide their age (14%). Two participants did not provide their post-tax monthly income (0.80%). The expectation-maximization (EM) algorithm in SPSS was used to impute the missing values.

### 2.2. Results

We conducted preliminary analysis, common method bias test, and hierarchical regression analysis using SPSS 22.0.

The results of Harman’s single-factor test showed that the most influential factor accounted for 22.54% of the total variance, which was below the 40% threshold [38]. Thus, the common method bias effect was not a serious problem in the current study. Table 1 shows the descriptive statistics and correlations among the variables.

We conducted a hierarchical regression analysis to test the effect of RD on ACS endorsement. Gender, age, post-tax monthly income, and educational level were entered in Step 1 and relative deprivation was entered in Step 2. The results show that after controlling for the demographic variables, RD positively predicted ACS endorsement, *β* = 0.41, *t*_200_ = 6.34, *p* < 0.001, 95% CI = [0.28, 0.54], Δ*R*^2^ = 0.15 (*f*^2^ = 0.18) (see Table 2). Hypothesis 1 was supported.

## 3. Study 2

Study 1 was correlational, which limits causal interpretation. Study 2 aimed to examine whether the manipulation of RD would influence ACS endorsement. In addition, Study 2 also investigated the mediating role of ACS endorsement.

### 3.1. Methods

#### 3.1.1. Participants and Procedure

For the main effect of relative deprivation on ACS endorsement, a priori power analysis was conducted using G*Power [35] and found that a minimum of 140 participants are needed to provide 90% power to detect an effect of Cohen’s *d* = 0.50, assuming a two-tailed test and α = 0.05 with an allocation ratio (N2/N1) = 1. For the mediating effect of relative deprivation, we used the Monte Carlo Power Analysis for Indirect Effect (https://schoemanna.shinyapps.io/mc_power_med/, accessed on 18 June 2019) [39] to determine the sample size. The results suggest that 301 participants are needed to have 90% power to detect the mediating effect of relative deprivation, assuming the correlation between RD and self-handicapping is 0.35, 0.30 between self-handicapping and ACS endorsement, and 0.37 between RD and ACS endorsement, and SDs are 1. Therefore, we aimed to recruit 301 participants at least. Finally, a total of 312 participants (136 males and 176 females) were recruited using online and offline posters and passed the attention check items. In the measures of Study 2, we inserted two attention check items as we did in Study 1. There were 47 participants who failed to pass the attention check items. As a result, data from 312 participants (136 males and 176 females) were entered into the final analyses. These participants ranged in age from 18 to 55 years old (Mage = 27.61, SD = 6.30). When they arrived at the lab, at the time they signed up for, participants signed a consent form. After the manipulation of RD, they completed the measures of self-handicapping tendency and ACS endorsement. Finally, participants received 10 RMB for their participation and were debriefed and thanked.

Manipulation of Relative Deprivation. Upon arrival, participants were randomly assigned to one of the two conditions: the RD condition (N = 156) and the control condition (N = 156). Following the manipulation in Sim et al. (2018), each participant was asked to imagine that they were working at a medium-sized company and that there were 15 people in their department [40]. At the end of each year, they were expected to receive a year-end bonus. In the RD condition, participants were told that not every employee would receive the same amount of bonus this year. Without any explanation, the participant received CNY 8500, while the other 14 coworkers in the same department received CNY 15,000. However, in the control condition, the participants were told that each employee received the same bonus this year, which was CNY 8500.

Following the manipulation, participants completed a two-item manipulation check. The items were “How satisfied are you with your bonus?” and “How fair do you feel you were treated?” [12,40]. These two items were rated from 1 (absolutely dissatisfied/unfair) to 5 (absolutely satisfied/fair).

#### 3.1.2. Measures

Self-Handicapping. Self-handicapping was evaluated using 16 items adapted from the Self-handicapping Scale [41]. Given that the original scale was designed to measure participants’ general level of self-handicapping, items were modified accordingly to measure participants’ situational self-handicapping after the manipulation. All items were rated on a 5-point scale from 1 = strongly disagree to 5 = strongly agree. The item scores were averaged, with higher scores indicating a higher level of self-handicapping (α = 0.73).

Anti-Chicken Soup Endorsement. To measure ACS endorsement, we composed five stories in which the protagonists are all experiencing hard times. As a result, the protagonists come up with some ACS ideology. The five stories were developed based on interviews with college students, asking them to tell us some difficulties or failures they or their friends had experienced in their lives. Finally, five commonly experienced difficult situations were selected. The ACS mottos held by the protagonists were all from the list in Study 1. Here is one example.

Wang was a high school student. She had studied hard to go to a top-notch university. However, she received only one offer from a second-tier university this year while many of her classmates who were similar to her were admitted to first-rate universities. Wang’s parents believe that hard work pays off. Thus, they suggested Wang study hard and take the examination one more time next year. However, Wang does not think that way. She believes that some people always fail no matter how hard they try, and the anti-chicken soup motto lingers in her ears: “nothing is difficult for the man who will give up”.

After reading each story, participants were asked to indicate whether they endorsed the protagonist’s belief and attitude on a 5-point Likert scale, from 1 (strongly disagree) to 5 (strongly agree). Scores on the five stories were averaged, with higher scores indicating a higher level of ACS endorsement. We believe that participants in the RD manipulation condition were likely to project their RD onto the protagonists in the stories and tended to agree with their ACS ideology. The reliability of the five-story measure was satisfactory (α = 0.81).

### 3.2. Results

As expected, participants in the RD condition (*M* = 1.87, *SD* = 0.87) reported a lower level of satisfaction (*t*_352_ = −23.33, *p* < 0.001, Cohen’s *d* = 0.80) than those in the control condition (*M* = 4.17, *SD* = 0.87). In addition, participants in the RD condition (*M* = 1.87, *SD* = 0.85) also reported a lower level of perceived fairness (*t*_352_ = −22.13, *p* < 0.001, Cohen’s *d* = 0.78) than those in the control condition (*M* = 4.10, *SD* = 0.93). These results suggest that the manipulation was successful.

The results of an independent samples t-test showed that participants in the RD condition (*M* = 3.58, *SD* = 0.59) reported a significantly higher level of ACS endorsement than those in the control condition (*M* = 2.08, *SD* = 0.59), *t*_352_ = 22.32, *p* < 0.001, Cohen’s *d* = 0.79, supporting Hypothesis 1.

A mediation model was tested using the bootstrapping technique (with 1000 iterations) [42] and Mplus 7.4 statistical software to determine the mediating effect of self-handicapping. The results (see Figure 1) indicate that the mediating effect of self-handicapping was significant (*β* = 0.21, *p* < 0.001, 95% CI = [0.14, 0.29]), thus supporting Hypothesis 2.

## 4. Discussion

In recent years, ACS has permeated people’s daily communication, entertainment, catering, and many other aspects. Because it advocates passive and negative attitudes toward hard work, making an effort [1], and emphasizes the disappointing aspects of life [4], this perspective on life may adversely affect people’s judgment of the world and themselves [5] which may be detrimental to their daily life and work [6]. Thus, it is important to explore the roots of such a social phenomenon. As one of the first empirical studies on ACS, the current study aims to test the relationship between RD and ACS endorsement as well as the underlying mechanism. The results of the current study support our hypotheses that people with high measured or manipulated RD are more likely to endorse ACS and that this effect is mediated by self-handicapping.

First, the current study confirms that RD is a key reason that could explain the popularity of ACS in society. The social conditions characterized by high economic inequality and low social mobility make RD a widespread feeling. In addition, the boom in social networking fosters the effect by amplifying the gap between rich and poor and making social comparisons at one’s fingertips. All these factors jointly make people’s sense of RD reach an unprecedented high level. Previous studies have found that RD positively predicts negative internal states, such as pressure, anxiety, depression, despair, and pessimism [7,19,20]. Similarly, ACS reflects a state of anxiety, hopelessness, desperation, discouragement, dissatisfaction, and depression [18]. Thus, the results of the current study are consistent with the extant literature and contribute to the theoretical development of RD by expanding its potential outcomes.

RD leads to a positive attitude toward ACS, which is generally pessimistic in nature. It seems paradoxical that the research also supports that RD is strongly associated with many risky behaviors (e.g., gambling) [12,37,43]. In fact, the targets of these two kinds of attitudes and behaviors are different. ACS is generally concerned with one’s focus on work and life. Engaging in risky behaviors means one is generally deviating from normal life and is off-track. Thus, it is reasonable to expect that when one cannot get what they want from their regular ways and feels despair and hopelessness, it is likely that they would engage in some risky behaviors to satisfy their desires.

The mediating effect of self-handicapping is crucial for us to understand why RD leads to ACS endorsement. When people are in a comparatively disadvantaged position, which they believe they should not be in, RD is likely to occur. Feeling relatively deprived is annoying, especially when people notice that their situation is difficult to change. In such conditions, people tend to rationalize their disadvantages without harming their self-worth and self-evaluation [22]. Thus, they are likely to adopt self-handicapping strategies [23] so that they could attribute their disadvantaged condition to external factors. Although self-handicapping temporarily protects self-esteem, it is costly: it lowers people’s motivation to make an effort to change the situation and leads to their endorsement of ACS, which precisely advocates the meaninglessness of working hard.

The findings of the current study also have important policy implications. ACS has gradually developed into a widely disseminated subculture. People are wearing T-shirts imprinted with the mottos, drinking coffee and tea with these mottos on the cups, posting such pictures and emojis on their WeChat moments, and bantering with each other using those mottos in their daily conversation. Many famous businesses choose to release unconventional advertising copies embracing ACS for the purpose of catering to their customers. Such a subculture might threaten the mainstream culture, which always encourages people to work hard. To curb the potential detrimental impact of the endorsement of ACS, it is important to understand why it happens. However, the extant literature is generally only theoretical speculation [1,5,18], and no empirical study has been done thus far. The current study takes the first step to understanding, from the RD perspective, why people abandon CS and embrace ACS instead. The findings of the current study suggest that if severe economic inequality and the accompanying feeling of RD are not relieved, it will be difficult to thoroughly address the subculture of ACS.

### Limitations and Future Directions

First, future studies could be performed in other social backgrounds to provide evidence of the generalizability of the findings from the current study, which was conducted in China. In fact, ACS is also popular in many other countries. For example, Pepe the Frog, or the “sad frog meme”, is an American cartoon drawing of a depressed-looking frog with a drooping mouth and a pair of ready-to-cry eyes, often accompanied by the text “You will never …” or “Feels bad man.” Its popularity has steadily grown since its creation in 2005. By 2015, it had become one of the most popular memes used on Tumblr and 4chan. On the other hand, economic inequality is also now a serious problem in the United States, and wealth controlled by the richest 0.1% is equal to that of the bottom 90% [44]. Studies based on participants from different societies, such as the United States, are welcomed to corroborate the model constructed in the current study.

Second, as the first attempt to empirically study a new social phenomenon, the current study develops some methods to measure ACS endorsement with satisfactory reliability. Specifically, we collect the most popular and typical ACS mottos in China, and we construct life stories advocating ACS ideology based on interviews with people in society. Then, participants’ attitudes are collected as an index of ACS endorsement. The merits of such measures are that they are vivid and tailored to people’s daily lives in specific social backgrounds. The shortcoming is that different materials have to be collected and constructed when measuring the construct in different societies, which makes the cross-cultural comparison of research results difficult. Further studies are necessary to develop an ACS endorsement scale. For example, a prototype analysis [45] would be performed to identify those features that people see as representative of ACS. With the cognitive and emotional representation people have of ACS, a general scale of ACS endorsement could be constructed.

Third, the current study constructs a mediation model to understand the relationship between RD and ACS endorsement. Further research is necessary to explore the potential boundary conditions of the findings. For example, RD threatens self-esteem, and self-handicapping was adopted as a coping strategy to protect self-esteem. It is possible that for people with high and stable self-esteem, the effect of RD on ACS endorsement via self-handicapping might be weaker because their self-esteem could serve as strong psychological strength to buffer the impact of disadvantaged situations. Future studies in these directions are strongly recommended.

Fourth, we manipulated RD and provided causal evidence of RD on self-handicapping and endorsement of ACS. However, the current study did not provide causal evidence of self-handicapping on ACS endorsement. Future studies are necessary to explore the causality by manipulating self-handicapping and to test its effects on ACS endorsement. In addition, to provide further evidence of the mediating effect of self-handicapping in the effect of RD the on the endorsement of ACS, a blockage manipulation design is needed, in which RD and self-handicapping will be manipulated concurrently [46]. Specifically, if RD affects the endorsement of ACS via self-handicapping, then the effect of RD on the endorsement of ACS should decrease or even vanish under the self-handicapping-blockage conditions (i.e., self-handicapping is manipulated to the same level across different levels of RD), compared with the self-handicapping-control conditions (i.e., self-handicapping is left to changes alongside with RD). Future studies are also needed to test the possibility of a reciprocal relationship between SH and ACS endorsement. That is, SH and ACS endorsement may mutually influence each other. Along with other negative consequences of ACS endorsement, endorsing ACS may also reinforce the motivation of self-handicappers to undermine their actions. Exploring the bi-directional relationship between the two can help us have a deeper understanding of the phenomenon of ACS endorsement.

## 5. Conclusions

The current studies were designed to explore the relationship between relative deprivation (RD) and the endorsement of anti-chicken soup (ACS) and the mediation role of self-handicapping. The results obtained reveal that measured (Study 1) and manipulated (Study 2) RD was associated with ACS endorsement and the relationship can be interpreted by self-handicapping. Relatively deprived people are likely to hinder their own motivation to try hard and attribute their failures or disadvantages to external reasons, which leads them to further endorse ACS.

## Figures and Tables

**Figure 1 ijerph-19-14210-f001:**
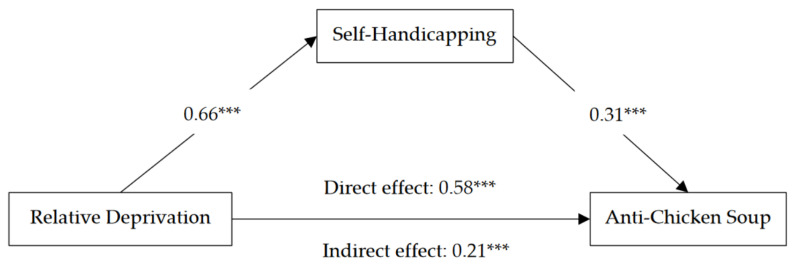
Indirect effect of relative deprivation on anti-chicken soup endorsement via self-handicapping in Study 2. Note. Relative deprivation (coded as 0 = control, 1 = relative deprivation). Values are standardized coefficients. *** *p* < 0.001.

**Table 1 ijerph-19-14210-t001:** Descriptive statistics and correlations among the variables.

	M	SD	1	2	3	4	5	6
1 Gender	-	-	-					
2 Age	27.61	6.30	0.159 *	-				
3 Income	1.76	0.91	0.257 ***	0.426 ***	-			
4 Education	3.04	0.59	−0.074	−0.204 **	−0.004	-		
5 RD	2.99	0.72	0.014	−0.314 ***	−0.147 *	−0.077	-	
6 ACS Endorsement	2.95	0.72	0.051	−0.039	−0.070	0.026	0.373 ***	-

Note. RD: relative deprivation; ACS: anti-chicken soup; * *p* < 0.05. ** *p* < 0.01. *** *p* < 0.001.

**Table 2 ijerph-19-14210-t002:** Hierarchical regression results for anti-chicken soup endorsement.

Variables	*β*	*B* (*SE*)	*t*	*p*	95% CI of *B*	
					LL	UL
Step 1						
Gender	0.08	0.12 (0.108)	1.12	0.262	−0.09	0.34
Age	−0.01	0.00 (0.009)	−0.11	0.910	−0.02	0.02
Income	−0.09	−0.07 (0.059)	−1.16	0.248	−0.19	0.05
Education	0.03	0.04 (0.083)	0.44	0.659	−0.13	0.20
Step 2						
Gender	0.05	0.08 (0.10)	0.81	0.421	−0.12	0.28
Age	0.13	0.02 (0.01)	1.85	0.066	0.00	0.03
Income	−0.08	−0.06 (0.06)	−1.15	0.250	−0.17	0.05
Education	0.09	0.11 (0.08)	1.40	0.162	−0.04	0.26
RD	0.41	0.41 (0.06)	6.34	<0.001	0.28	0.54
*R* ^2^	0.16					
Δ*R*^2^	0.15 ***					

Note. RD: relative deprivation; CI = confidence interval; LL = lower limit; UL = upper limit; *** *p* < 0.001.

## Data Availability

The data of the current study are available in the Mendeley Data repository Available online: http://dx.doi.org/10.17632/zcm6rnv9bb.2 (accessed on 25 January 2021).

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
