# Peer review of "Relative Deprivation Leads to the Endorsement of “Anti-Chicken Soup” in China"

_ijerph, 2022, doi:10.3390/ijerph192114210_

Round 1
Reviewer 1 Report
A very interesting study! Anti-chicken soup is very popular among young people in Mainland China today. The current study aims to find the reason why people endorse the anti-chicken soup. The questionnaire study and the manipulation study were well designed and supported the authors’ hypotheses. However, the current study still has room to improve.
Firstly, the DV of the study is the endorsement of anti-chicken soup. But I want to know why people can not endorse ACS. Maybe it’s not a bad choice for those who are under economic inequality and RD. Otherwise, maybe ACS can make them relax and feel less pressure, just like humor and one laugh at oneself. Maybe it’s their wisdom to cope with the tough situation. So, maybe the authors need to point out the severe results of endorsing the ACS, then the current study can regain its significance.
Secondly, I don’t know why self-handicapping could lead to the endorsement of ACS. It seems that both of them are the result of RD. In the introduction part, the authors may need to explain why it is self-handicapping leading to ACS, rather than ACS leading to self-handicapping. And when running the mediation model, the author may also need to run the alternative models, like RD→ACS→self-handicapping. I also suggest that the author verify the mediation model by manipulating the self-handicapping. If the SH is the mediator, then we may expect that if participants are given the chance of not handicapping, then they would not endorse ACS.
Thirdly, there are minor errors in the model. The DV should be the endorsement of ACS, rather than ACS. ACS is not a variable.
Reviewer 2 Report
The article focused on an interesting social phenomenon(Anti-Chicken Soup)from the perspective of relative deprivation, and which was conducted with cross-sectional method and experimental design. Overall, the topic of the article is interesting and the method is standardized. The following comments of the manuscript can be considered for its revision.
1. The logic of the article is: Study 1 confirms the relationship between relative deprivation and Anti-Chicken Soup Endorsement, and Study 2 confirms the mediating role of self-handicapping.
However, in the introduction, evidence of the hypothesis 1(RD is positively associated with people’s endorsement of ACS.) is based on hypothesis 2 (the mediating effect of self- handicapping). I am confused about this. The evidence of hypothesis 1 should be added, and the logic of the hypothesis in the article needs to be clarified.
2. The basis for the number of participants in Study 2 is not presented in the Methods section.
Round 2
Reviewer 1 Report
The quality of the paper has been greatly improved. Well, one more thing to be improved is the authors should report the alternative model of RD→the endorsement of ACS→Self Handicapping. After all, theoretical reasoning alone can not exclude alternative explanations.
